# Dietary Supplementation with Chlorogenic Acid Enhances Antioxidant Capacity, Which Promotes Growth, Jejunum Barrier Function, and Cecum Microbiota in Broilers under High Stocking Density Stress

**DOI:** 10.3390/ani13020303

**Published:** 2023-01-15

**Authors:** Yanhao Liu, Yi Zhang, Dongying Bai, Yuqian Li, Xianglong He, Koichi Ito, Kexin Liu, Haiqiu Tan, Wenrui Zhen, Bingkun Zhang, Yanbo Ma

**Affiliations:** 1Department of Animal Physiology, College of Animal Science and Technology, Henan University of Science and Technology, Luoyang 471003, China; 2Henan International Joint Laboratory of Animal Welfare and Health Breeding, College of Animal Science and Technology, Henan University of Science and Technology, Luoyang 471023, China; 3Department of Food and Physiological Models, Graduate School of Agricultural and Life Sciences, The University of Tokyo, Ibaraki 319-0206, Japan; 4State Key Laboratory of Animal Nutrition, Department of Animal Nutrition and Feed Science, College of Animal Science and Technology, China Agricultural University, Beijing 100193, China; 5Innovative Research Team of Livestock Intelligent Breeding and Equipment, Longmen Laboratory, Luoyang 471023, China

**Keywords:** high stocking density stress, chlorogenic acid, antioxidant capacity, intestinal barrier function, microbial community, broilers

## Abstract

**Simple Summary:**

High density (HD) rearing seriously hinders the healthy development of broilers. Chlorogenic acid, a natural antioxidant food and drug source, may relieve the HD oxidative stress. Therefore, the present study was to investigate the effects of CGA supplementation on growth performance, antioxidant index in serum, relative mRNA expression of jejunum, and cecum microflora of broilers expose to HD stress. A total of 476 7-day-old AA+ broilers were divided into normal density (ND group, 14 broilers/m^2^) and high density (HD group, 22 broilers/m^2^), and CGA supplementation (ND or HD + CGA) group. The results showed that HD stress decreased the serum antioxidant capacity, inhibited the development of intestinal villi and the expression of tight junction genes, and thus affected the intestinal permeability during the rapid growth period from 21 to 35 days. Meanwhile, with the progression of HD oxidative stress, adverse changes occurred in intestinal microorganisms, leading to intestinal inflammation, and finally affected the growth performance on day 42. Additionally, dietary CGA can inhibit the oxidative stress from serum to intestinal tract, improve intestinal integrity, and enhance beneficial flora of HD broilers. Thus, dietary supplementations of an antioxidant such as CGA can effectively prevent oxidative stress in HD broilers.

**Abstract:**

Chlorogenic acids (CGA) are widely used as feed additives for their ability to improve growth performance and intestinal health in poultry. However, whether dietary CGAs could reverse the impaired intestinal condition caused by high stocking density (HD) in broiler chickens is unknown. We determined the effect of dietary CGA on growth, serum antioxidant levels, jejunum barrier function, and the microbial community in the cecum of broilers raised under normal (ND) or HD conditions. HD stress significantly decreased growth and body weight, which was restored by CGA. The HD group showed increased serum malondialdehyde, an oxidative byproduct, and decreased SOD and GSH-Px activity. CGA reduced malondialdehyde and restored antioxidant enzyme activity. HD stress also significantly decreased jejunal villus length and increased crypt depth. Compared with ND, the expression of tight-junction genes was significantly decreased in the HD group, but this decrease was reversed by CGA. HD also significantly upregulated TNF-α. Compared with ND, the cecal microbiota in the HD group showed lower alpha diversity with increases in the harmful bacteria *Turicibacter* and *Shigella*. This change was altered in the HD + CGA group, with enrichment of *Blautia*, *Akkermansia*, and other beneficial bacteria. These results demonstrated that HD stress decreased serum antioxidant capacity, inhibited the development of jejunal villi, and downregulated expression of tight-junction genes, which increased intestinal permeability during the rapid growth period (21 to 35 days). Dietary CGA enhanced antioxidant capacity, improved intestinal integrity, and enhanced beneficial gut bacteria in chickens raised under HD conditions.

## 1. Introduction

In the livestock industry, rapidly increasing attention is being paid to the problems of animal health and welfare, especially in poultry production where animals are often exposed to environmental stress factors that are strongly affected by stocking density [1,2,3]. Nevertheless, stocking density is an important management measure for poultry production and well-being [3]. High stocking density (HD) is often used to enhance profitability because it increases chicken production per fixed stocking area [4]. Normally, an extreme stocking density of 39 kg/m^2^ or 18 birds/m^2^ is acceptable as long as they are properly reared in windowless rooms [4]. However, to meet the growing demands of production, stocking densities much higher than the above density have been used in the poultry industry [5]. The resulting HD typically causes increased oxidative stress which in turn leads to low growth performance and weak immune function [6].

Previously, HD was related to abnormal broiler behavior in addition to problems such as dermatitis, poor feathering and soiling, and more importantly, increased oxidative damage [3,7]. Moreover, oxidative stress from HD could induce a harmful physiological states and oxidative damage from overproduction of free radicals, lowering of antioxidant defenses, and inflammation [8]. This condition can overwhelm the host’s defensive capacity and raises the level of oxidative stress products such as malondialdehyde in serum [7], which can lead to oxidative injury of jejunal tissues [9]. Specifically, HD stress can degrade the integrity of the intestinal barrier by affecting expression of the *claudin-1*(*CLDN1*), *occluding*(*OCLN*), and *zonula occludens-1* (*ZO-1*) genes [10], which were regularly accompanied by increased pro-inflammatory cytokines such as *interleukin-1β* (*IL-1β*) and *tumor necrosis factor-α* (*TNF-α*) [11]. Other researchers have also reported alterations in the intestinal microbial population of broilers in response to HD [12]. HD stressors significantly altered the alpha and beta diversity of the intestinal microbial community of broiler chickens [13]. However, whether damage to physiological and immune states induced by HD stress was associated with growth remained to be confirmed.

Alternatives to antibiotics have been used to enhance growth and antioxidant capacity under high stocking density in broilers [13], including traditional Chinese herbal formulations, probiotics, and antibacterial peptides [13,14,15]. Chlorogenic acid (CGA), an important bioactive dietary polyphenol, is esterified by quinic and caffeic acids, which are produced by green coffee bean extracts [16]. Several studies have reported that CGA has significant pharmacological properties as well as strong antioxidant activity [17]. CGA also had anti-inflammatory effects, including decreased *IL-1β*, *interleukin-6* (*IL-6*), and *TNF-α*, and increased *interleukin-10* (*IL-10*) [17]. Recent data have shown that CGA increases intestinal barrier integrity and immune function by modifying the gut microbial composition of broiler chickens [16,17]. In addition, under the current situation of banning antibiotics in feed and restricting livestock and poultry breeding to reduce drug resistance, natural plants with drug homology have become the future development direction for animal husbandry [18]. CGA is considered a potential feed additive to promote animal health and prevent antibiotic abuse [19]. Hence, this study aimed to explore whether dietary supplementation with CGA could improve growth performance, antioxidant metabolism, and related gene expression, and the microbiome of the jejunum and cecum in broilers exposed to HD stress.

## 2. Materials and Methods

### 2.1. Ethical Treatment

This project design received approval from the Care and Use of Experimental Animals Committee of the Henan University of Science and Technology (AW20602202-1-2) in 2019. Experiments were performed following defined guidelines of the Henan University of Science and Technology for animal care.

### 2.2. Animals and Experimental Design

A group of 476 healthy day-old male Arbor Acres broiler breeders were purchased from a commercial hatchery (Henan Quanda Poultry Breeding Co., Ltd., Hebi, China), and experiments were performed at the Animal Research Unit of Henan University of Science and Technology. During the initial week of the experiments, all animals were kept at 33 °C ± 1 °C. The temperature was then decreased by 1–2 °C every week, until it reached 25 °C ± 2 °C by day 42. The relative humidity was kept at 60–70%, the lights were on for 23 h every day, and off for one hour from 7 to 8 p.m. at night. On day 7, the chickens were weighed and randomly divided into four treatment groups: normal stocking density of 14 birds per m^2^ (ND), high stocking density of 22 birds per m^2^ (HD), and dietary supplementation with CGA at 1.5 g/kg in the ND and HD groups (ND + CGA and HD + CGA). CGA (purity 98%) was purchased from Changsha Staherb natural ingredients Co., Ltd. (Changsha, China) and directly added this concentration to the base diet. Each of the treatments consisted of five replicates and three preparations. The initial assessment of a focus trial showed that a 0.15% CGA dosage yielded a better outcome on HD stress of broilers than 0.1% and 0.2% CGA. Thus, it was determined that the CGA concentration should be 0.15%. A diet of corn and soybean meal was fed to the animals following NRC guidelines [20] (Table 1) with no additives except an anticoccidial treatment. Birds were maintained at constant temperature and humidity with food and water ad libitum. Animals in all of the five treatment groups were given vaccinations for infectious bronchitis virus and Newcastle disease virus on days 1 and 20, and bursal disease virus on day 14, as previously described [21].

### 2.3. Growth Performance and Sample Collection

At 42 days of age, the body weight (BW) and average daily gain (ADG) of birds in each cage were recorded after 8 h of fasting. Two animals from each treatment group were randomly picked and euthanized by CO_2_ inhalation for sampling jejunal tissue. Blood was drawn from wing veins and placed into serum separator tubes (SST; BD Bioscience, Franklin Lakes, NJ, USA), and then centrifuged at 1409 g for 10 min to obtain serum, which was held at −20 °C for later assay. A 2-cm segment of the middle part of the jejunum was removed, snap frozen in liquid N_2_, and the cecum digesta were collected and stored at −80 °C. The whole sampling process was completed in 25 min.

### 2.4. Antioxidant Capacity of the Serum

Serum aliquots (100 µL) were analyzed for malondialdehyde (MDA) and catalase (CAT) content and glutathione peroxidase (GSH-Px) and superoxide dismutase (SOD) activity (kits A001-3, A007-1-1, A005-1 and A001-3, respectively; Nanjing Jiancheng Bioengineering Institute, Nanjing, China.

### 2.5. Intestinal Morphology

Hematoxylin and eosin (H&E) staining was used to evaluate intestinal histology. The jejunum tissue samples were fixed in 4% paraformaldehyde, dehydrated in a succession of increasing ethanol treatments, cleared with xylene, and paraffin-embedded. The wax blocks were cooled to −10 °C, sectioned, and held at RT. The sections were stained with hematoxylin for 8 min, differentiated with 1% HCl in ethanol for 10 s, and washed with DW for 10–15 min. Sections were then counterstained with eosin for 1 min, washed with DW for 6 min, dehydrated with a graded sequence of ethanol consequences, anhydrous ethanol and xylene, then cover-slipped and dried overnight. The tissue sections were observed under the microscope (200×) and the Servicebio system was used to collect images. The heights of the villi and depths of the crypts of the jejunum were measured with the microscope image analysis software C.V2.0, and ratios of villus height/crypt depth (VH/CD) were determined.

### 2.6. Quantiative Real-Time PCR (qRT-PCR) Analysis

RNA extraction and real-time PCR were performed as described in previous study [22]. Briefly, total RNA was isolated was isolated from the jejunal tissues with TRIzol reagent (Invitrogen, Carlsbad, CA, USA) according to the manufacturer’s instructions, and the purity and concentration of RNA were measured spectrophotometrically (Nano-drop2000). An A_260_/A_280_ ratio of 1.9–2.0 indicated good quality RNA, usable for the next step. cDNA was synthesized by employing a Primer Script™ RT reagent Kit with a gDNA Eraser (Takara Biotechnology Co., Ltd., Tokyo, Japan). Quantitative real-time PCR was executed following the protocol reported by [22]. Fluorescent quantitative PCR was performed with target gene-specific primers designed with NCBI/Primer-BLAST and primer specificity was confirmed [23]. The primers for the jejunal tight-junction genes and immune-specific genes are given in Table 2. The relative gene expression levels were determined using the 2^−∆∆CT^ method, and *GAPDH* was employed as the reference gene for normalization.

### 2.7. The Cecal Bacterial Community

Total microbial DNA was extracted from cecal contents (100 mg) using the QIAamp Fast Stool Mini Kit (Qiagen, Hilden, Germany) and stored at −80 °C. The V3–V4 region of the 16S rRNA gene was amplified with the universal primers, F341 and R806: F: ACTCCTACGGGAGGCAGCA; R: GGACTACHVGGGTWTCTAAT. A Qiagen gel-extraction kit (Qiagen, Germany) was used to extract the PCR amplicons, which were quantitated with a Qubit 2.0 fluorometer (Thermo-Fisher, Waltham, MA, USA) and pooled. Amplicon libraries were sequenced on the Illumina MiSeq PE250 platform (Illumina, San Diego, CA, USA). The sequencing was performed by Personal Biotechnology, Co., Ltd. (Shanghai, China) and data were analyzed using the free online platform, Personalbio GenesCloud

(https://www.genescloud.cn, accessed on 17 August 2022). Analysis of MiSeq sequencing data was based on a previous report [22]. Briefly, A sample of cecal material was submitted to Shanghai Personal Bio-technology Co., Ltd. (Shang hai, China) to be analyzed for 16S rRNA. The Illumina platform was used to carry out the processing of the raw sequenc-ing data. Using the DADA2 method, sequences of high quality were clus-tered into Operational Taxonomic Units (OTUs) with a 97% similarity cutoff, involving depriming, quality filtering, denoising, splicing, and dechimera steps. The OTUs were evaluated and labeled based on the Greengenes da-tabase. The extraction and overlapping of OTUs, clustering, α-diversity, and β-diversity were obtained by means of the QIIME2 (2019.4) software with Python scripts. A species-diversity matrix was prepared based on binary Jaccard, Bray–Curtis, and weighted and unweighted UniFrac algorithms. The linear discriminant analysis (LDA) effect size (LEfSe) was employed to determine the differential abundance of taxa. The UPGMA clustering heatmap of microbial communities was analyzed among the top 20 genera based on the abundance. The sample FASTQ information was submitted to the NCBI (Samples accession number: SRP414903-PRJNA916212).

### 2.8. Statistical Analyses

Each cage-replicate was defined as an experimental unit for growth-performance assessment. The individual animals in each replicate were used as the experimental units (*n* = 10) for other analyses. The data were compared by one-factor ANOVA using SPSS20.0 (SPSS Inc., Chicago, IL, USA). Differences between groups were evaluated by means of Tukey’s multiple-range test and were considered significant at *p* < 0.05. The figures were constructed with GraphPad Prism 9 (GraphPad Software Inc., San Diego, CA, USA).

## 3. Results

### 3.1. Effect of CGA on Growth Performance of HD Broilers

The effect of CGA on the growth of HD chickens is shown in Figure 1. Relative to the ND group, the BW in the HD group on day 42 was significantly lower (*p* < 0.05), and dietary supplementation with CGA also significantly increased (*p* < 0.05) the BW on the same day in the ND + CGA and HD + CGA groups. In addition, the ADG was significantly increased (*p* < 0.05) in the ND + CGA group relative to ND, while the ADG in the HD + CGA group was higher (*p* < 0.05) than that in the HD group.

### 3.2. Effect of CGA on Antioxidant Capacity of HD Broilers

The serum antioxidant indices of broilers in each treatment group are shown in Table 3. Relative to ND, the serum content of MDA was higher (*p* < 0.05) in the HD group on days 28, 35, and 42. Dietary supplementations with CGA significantly decreased (*p* < 0.05) MDA in the HD + CGA compared to the HD group on the same days. In addition, the serum SOD level in HD group was significantly lower than that in ND group at 28 and 42 days (*p* < 0.05) and GSH-Px level in HD group was significantly lower than that in ND group at 21 and 28 days (*p* < 0.05). However, the SOD and GSH-Px level was higher (*p* < 0.05) in the HD + CGA group than in the HD group on days 28, 35 and 42. Relative to the ND group, the CAT content was higher (*p* < 0.05) in the HD + CGA group on day 42, relative to the HD group, the CAT content was higher (*p* < 0.05) in the HD + CGA group on day 42.

### 3.3. Effect of CGA on Jejunum Morphology in HD Broilers

The height of the villi, the crypt depth, and the VH/CD ratio are shown in Figure 2 and Table 4. Relative to the ND group, the villus height and VH/CD were significant lower (*p* < 0.01) in the HD group on days 21 and 28, and higher in the HD + CGA group (*p* < 0.05) than the HD group. The crypt depth in the HD group was greater (*p* < 0.05) than the ND group on days 21 and 42. On day 28, the crypt depth of ND group was higher than that of ND + CGA group (*p* < 0.05), but there was no significant difference between the HD and HD + CGA groups on day 35.

### 3.4. Effect of CGA on Jejunal mRNA Expression of Tight-Junction Genes and Immune Factors in HD Broilers

The relative mRNA expression in the jejunum of each treatment group is shown in Figure 3 and Figure 4. Expression of the tight-junction genes, *OCLN*, *CLDN-1*, *CLDN-2*, and *ZO-1*, was significantly decreased (*p* < 0.05) in the HD group relative to the ND group on days 21 and 28. These mRNAs were significantly upregulated (*p* < 0.05) by dietary supplementation with CGA. The expression of *CLDN-2* and *ZO-1* were significantly decreased (*p* < 0.05) in the HD group compared to ND, and the HD + CGA group showed significant upregulation (*p* < 0.05) of these genes on days 35 and 42. Compared with ND, the expression of *TNF-α* was significantly increased (*p* < 0.05) in HD chickens on days 21 and 28, while the HD + CGA group showed significant downregulation (*p* < 0.05) of *TNF-α* expression on days 21 and 28. Additionally, the expression of *IL-1β* and *IL-6* was significantly increased (*p* < 0.05) in the HD group compared with ND on days 28, 35, and 42, and downregulated in the HD + CGA group (*p* < 0.05) compared to untreated HD chickens. The expression of *IL-10* was significantly increased (*p* < 0.05) in the HD + CGA group relative to the HD group on day 35.

### 3.5. Effect of CGA on the Composition and Diversity of Cecal Microbial Flora of HD Broilers

A significant clustering of microbes in the cecal microbiota was observed between ND, HD and HD + CGA group (Figure 5). In addition, the Shannon, Simpson, and Chao bacterial richness and alpha-diversity indices of the HD group were lower (*p* < 0.05) than for the ND group, while the HD + CGA group had significantly increased Shannon and Simpson indices (*p* < 0.05) (Figure 5A–C). The principal coordinate analysis (PCoA) (Figure 5D) revealed that the HD and HD + CGA groups had relatively fragmented distributions with low similarity and large differences in community composition, while the ND group showed a relatively concentrated distribution, differing from that of the other two groups. As shown in the Venn diagram analysis of the OTUs, there were 1147 share not duplicated OTUs among the three groups; 7524, 3359, and 3652 unique OTUs were identified in the ND, HD, and HD + CGA groups, respectively (Figure 5E). Taxonomic unit analysis revealed that the dominant phyla (Figure 5F) of the three groups included Firmicutes (77.98%, 88.92%, 88.92%), Bacteroidetes (19.73%, 7.64%, 10.14%), and Actinobacteria (0.63%, 0.96%, 0.73%). The HD and HD + CGA groups had higher numbers of Firmicutes and lower numbers of Bacteroidetes compared with the ND group (*p* < 0.05). There was no significant difference in the proportion of Actinobacteria between HD and HD + CGA. At the genus level (Figure 5G), the dominant genera in the ceca of the three groups were *Lactobacillus* (24.98%, 27.28%, 30.24%), *Bacteroides* (16.33%, 5.32%, 5.77%), *Faecalibacterium* (7.46%, 3.87%, 5.52%), *Ruminococcus* (2.24%, 4.62%, 2.86%), *Alistipes* (3.13%, 2.21%, 4.33%), *Oscillospira* (3.82%, 2.71%, 2.81%), and *Butyricoccus* (2.54%, 2.66%, 1.93%). Compared with ND group, the HD group had fewer (*p* < 0.05) *Faecalibacterium* and *Alistipes* (Figure 5H), but there was no difference among the three groups with regard to the other dominant bacterial genera. The LEfSe results (Figure 5I,J) showed that Actinobacteria, Bifidobacteriales, Bifidobacteriaceae, and Actinomycetales were enriched in the ND group compared to the HD group. Lactobacillaceae, *Lactobacillus*, Lactobacillales, *Weissella*, *Abiotrophia*, and Aerococcaceae were enriched in the HD + CGA group compared to the HD group.

To further compare the differences in species composition among groups and to achieve a demonstration of species abundance distribution trends for each sample, the top 20 genera in terms of relative abundance were clustered at the genus classification level and plotted on a heat map (Figure 5K). *Alistipes*, *Anaerotruncus*, and *Faecalibacterium* were greatly enriched in the ND group, *Turicibacter* and *Shigella* were greatly enriched in the HD group, and the relative abundance of *Cc-15*, *Akkermansia*, and *Blautia* was increased in HD + CGA.

## 4. Discussion

Many previous experiments have reported that chickens raised under high density conditions had lower BWs and ADG than those in normal density [24,25]. Additionally, some studies showed that increasing the stocking density increased stress and led to reductions in BWG during the growing periods [2], which was in agreement with our current study. Esteves [26] claimed that the recommended stocking density could be increased to nearly 18 birds/m^2^ with no major adverse effects on the final BW. The HD conditions in our study were over the recommended density, demonstrating that HD negatively affected weight gain and feed intake. CGA supplementation increased BW and ADG of HD chickens, suggesting that addition of CGA to the diet could alleviate the anorexia induced by HD housing, similar to the results of the previous study [21], although a dose of 1 g/kg (0.1% CGA) was administered.

In the present study, HD increased MDA levels on days 21 and 42, and decreased the antioxidant enzymes SOD, GSH-Px, and CAT on days 28 and 42, which are the key intracellular defenses to oxidative stress in broilers [27]. These results suggested that HD induced oxidative stress during the fast-growing periods, which was consistent with the growth performance results. Moreover, dietary CGA supplementations decreased MDA levels on days 21 and 42, increased SOD on days 21 and 42, and GSH-Px on days 28 and 35 in the serum of HD chickens. Our results demonstrated that feeding CGA effectively alleviated HD-induced oxidative stress by enhancing antioxidant enzyme activity in chickens. These results are similar to those in the study by [28], who found that antioxidant enzymes activities were enhanced with dietary supplementations of CGA in heat-stressed chickens.

Oxidative stress is associated with many intestinal disorders and their pathological processes [29]. Intestinal morphology, villus height, crypt depth, and V/C ratio are key indicators of gut health, which can reflect the intestinal barrier integrity, development status, and the nutrient absorption capacity of the intestine in chickens [30]. In this study, our results showed that HD housing of chickens significantly injured the intestinal morphology in the jejuna, similar to results in previous work [31,32]. In addition, we showed that CGA feeding also increased villus height and V/C value and decreased crypt depth of the ileum in chickens under oxidant stress. The improvement in gut health and antioxidant enzymes in our experiments may account for the improvement in growth. Recent reports revealed that CGA supplementations improved intestinal morphology in chickens with necrotizing enteritis [31]. Meanwhile, Liu [30] found that dietary supplementation with 1 g/kg CGA increased villus height and maintained intestinal integrity of chickens with necrotic enteritis.

Normal intestinal morphology and intact intestinal barrier are essential for normal metabolism and immunocompetence [33,34]. Tight junction genes such as *OCLN*, *CLDN-1*, *CLDN-2*, and *ZO-1*, constitute a mechanical barrier that is particularly essential for gut health in chickens [35]. Our result suggested that on days 21 and 28, HD resulted in the downregulation of mRNA expression of tight-junction-related genes in the jejunum, which constitutes evidence of impaired intestinal barrier defenses [36]. *OCLN* and *CLDN-1* are transmembrane tight-junction genes, which are involved in preserving tight junctions and the gut barrier [37]. *ZO-1* is an essential scaffolding protein connecting transmembrane proteins to intracellular cytoskeletons [37]. In addition, our data suggested that supplementation of HD chickens with CGA significantly increased the expression of tight-junction-related genes in jejuna thereby improving intestinal barrier function. Similar results were found in a study by Liu [30], who reported that dietary CGA upregulated the expression of tight junction proteins in dexamethasone-challenged chickens. In summary, combined with the histomorphology results and tight-junction gene expression, it is suggested that in the early rapid growth stage, HD caused damage to the development of intestinal villi and the intestinal barrier, ultimately reducing production performance of the whole cycle.

Cytokines play a critical role in immune and inflammatory responses [38]. Exposure to HD conditions increased the expression of pro-inflammatory factors in broilers [39,40]; inflammation is one of the main factors in the disruption of intestinal barrier function, and numerous studies have shown that upregulation of *TNF-α*, *IL-1β*, and *IL-6* and downregulation of *IL-10* underlie damage to the intestinal barrier [41,42,43]; CGA reduced excessive *TNF-α* and *IL-1β* expression and significantly inhibited inflammation [44]. In this study, the expression of *TNF-α*, *IL-1β*, and *IL-6* in jejunal tissue was significantly increased in the HD group compared with ND chickens, while expression of *TNF-α*, *IL-1β*, and *IL-6* tended to decrease with CGA feeding. Although there were differences in the expression of these genes throughout the dietary supplementation period, they were active during the rapid growth period after 28 days of age. There have been reports of research suggesting that the response to changes in stocking density depended on environmental variables [45]. Especially during rapid growth periods, high stocking densities may result in decreased broiler performance due to high temperatures and low airflow [3]. Dawkins and colleagues [46] proposed that environmental differences in the poultry producers’ methods of rearing chickens may have a greater impact on welfare than stocking density. In the present study, on days 35 and 42, the mRNA expression of *IL-10* in the ND + CGA group was significantly increased over that of the treatment HD group, and this effect was reversed at day 28. *IL-10* mainly targets the innate and adaptive immune responses and exerts an immunosuppressive function to reduce excessive and uncontrolled inflammatory responses. This over-activity of the immune system can cause tissue damage, especially during the remission phase of infection and inflammation, and has been suggested as a target in inflammatory diseases to restore tissue homeostasis [47]. The results suggest that with the growth of chickens, the HD increases stress, which leads to excessive inflammation, and then induces *IL-10* overexpression. Dietary supplementations with CGA could inhibit the occurrence of excessive inflammation.

It is well known that the intestinal microbiota of poultry plays an important role in digestion, barrier maintenance, and immune function, all of which may contribute to enhanced growth performance [48,49]. The intestinal microbial population is a complex community of hundreds of different microorganisms which affects the host and plays a role in regulating the immune system, nutrient digestion, and modulating intestinal function as well as overall health and growth [37,50]. It is clear from this study that HD could disrupt the balance of gut microbes, not only by reducing their diversity, but also by allowing the growth of pathogenic bacteria, which is consistent with previous research [51]. In the present study, dietary supplementation with CGA significantly increased the cecal microbial diversity in HD chickens as evidenced by the Shannon, Simpson, and Chao1 indices, likewise, the previous studies revealed that CGA not only significantly changed the intestinal structure, but also increased the α diversity of cecal microbes, decreased the relative abundance of Firmicutes and Proteobacteria in cecal digests, and increased the relative abundance of Bacteroidetes and the relative abundance of beneficial bacteria [52]. Moreover, in this study, the number of Firmicutes in the ND group was significantly lower than that in the HD group, indicating that the intestinal health of chickens in the HD group was damaged. The same studies showed that at the phylum level, the intestinal microbiota was mainly composed of Firmicutes (35–80%) [53]. Elevated corticosterone levels induced by high density stress may lead to dysregulation of intestinal microbiota, including an increased abundance of Firmicutes and decrease in α diversity [25]. Lefse analysis showed that Actinobacteria and Bifidobacteriales were the dominant types in the ND group compared with the HD group, which has been shown to play a dominant role in the maintenance of intestinal health [54]. Compared with HD group, the HD + CGA group was enriched in *Lactobacillaceae*, *Lactobacillus*, Lactobacillales, and *Weissella*, and these bacteria have likewise been shown to play an important role in the maintenance of organismal health [55,56]. In this study, *Faecalibacterium* and *Alistipes* were abundantly enriched in the ND group, but less abundant in the HD group; the HD + CGA group had an increased relative abundance of these genera. Numerous studies demonstrated that *Clostridium pratense* (Faecalibacterium) and *Alistipes* play an important role in immune regulation and maintenance of intestinal homeostasis [57,58]. CGA supplementation improved the body’s defense against oxidative stress and inflammatory response damage, while the numbers of *Alistipes* also increased in the study [44].

Several studies have shown that *Ruminococcus* was associated with a variety of intestinal diseases [59,60], and *Shigella* was one of the main pathogens causing intestinal infections [61,62]. These bacteria were significantly enriched in the HD group, and CGA reduced their abundance. These results suggested that CGA could also ameliorate HD stress by enhancing intestinal microbial activity. The cecal microbiome profile suggested that with exposure to HD, CGA may have positive effects such as increasing the abundance of beneficial microorganisms and decreasing the abundance of bacteria associated with dysfunction, thus reducing oxidative damage, maintaining immune regulation, and ultimately protecting the integrity of the intestinal barrier.

## 5. Conclusions

The present study preliminarily demonstrated that changes in the environment from HD stress during the rapid growth period from days 21 to 35 affected serum antioxidant levels, inhibited the development of intestinal villi, and thus affected intestinal permeability. Meanwhile, along with the progression of oxidative stress from HD, adverse changes occurred in the intestinal microbiota that led to intestinal inflammation, eventually reducing growth performance by day 42. However, dietary CGA supplementation could repress chronic inflammation, improve intestinal integrity, and enhance beneficial gut microbes in broilers challenged with HD stress. Overall, this study reveals a promising strategy for the prevention of chronic intestinal inflammation in high-stocking density-reared broilers by dietary supplementations with immunomodulators such as CGA.

## Figures and Tables

**Figure 1 animals-13-00303-f001:**
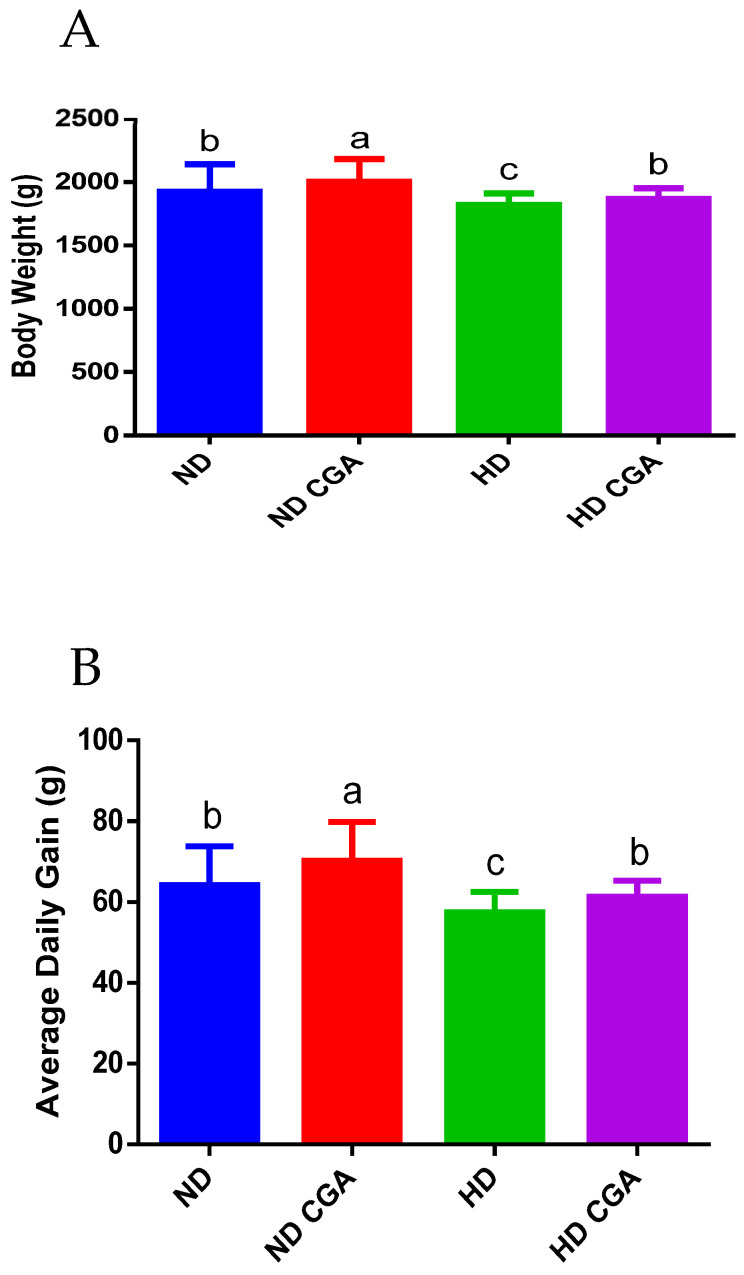
Effects of CGA on growth performance of broilers under HD stress. ND group, normal stocking density + basal diet; ND + CGA group, normal stocking density + basal diet + 0.15% CGA; HD group, high stocking density + basal diet; HD + CGA group, high stocking density + basal diet + 0.15% CGA. (**A**) Body weight at day 42. (**B**) Average daily gain at days 21 and 42. Each vertical bar represents the mean ± SEM (n = 10). Values with different letters are significantly different (*p* < 0.05, Tukey’s HSD test after one-way ANOVA).

**Figure 2 animals-13-00303-f002:**
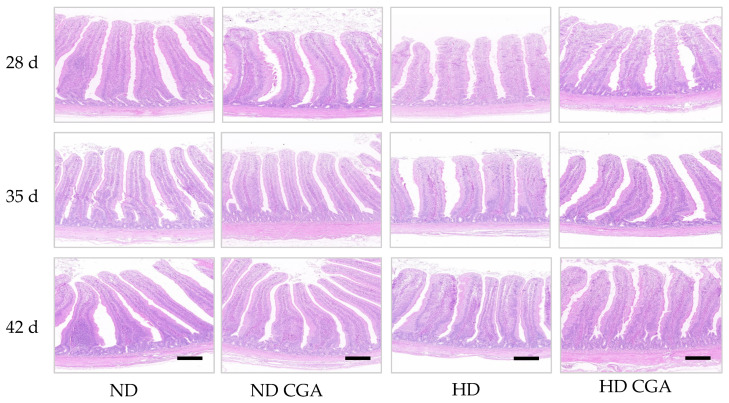
Effects of CGA on jejunal morphology of broilers under HD stress. ND group, normal stocking density + basal diet; ND + CGA group, normal stocking density + basal diet + 0.15% CGA; HD group, high stocking density + basal diet; HD + CGA group, high stocking density + basal diet + 0.15% CGA. Scale bar = 100 µm.

**Figure 3 animals-13-00303-f003:**
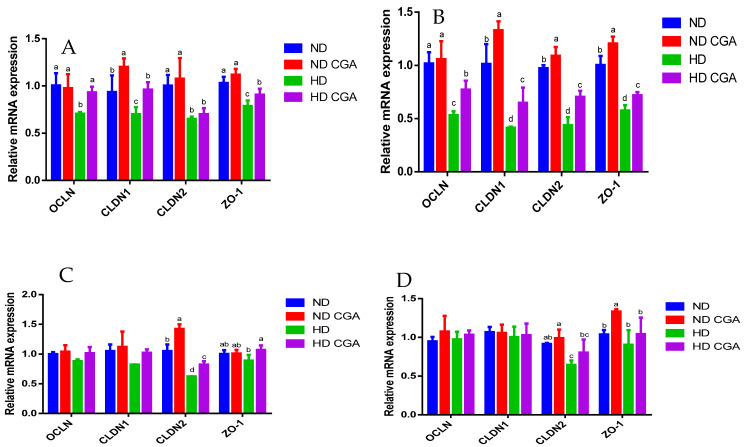
Effects of CGA on jejunal mRNA expression of tight junction genes in broilers under HD stress. ND group, normal stocking density + basal diet; ND + CGA group, normal stocking density + basal diet + 0.15% CGA; HD group, high stocking density + basal diet; HD + CGA group, high stocking density + basal diet + 0.15% CGA. (**A**) Relative mRNA expression at day 21. (**B**) Relative mRNA expression at day 28. (**C**) Relative mRNA expression at day 35. (**D**) Relative mRNA expression at day 42. Each vertical bar represents the mean ± SEM (n = 10). Values with different letters are significantly different (*p* < 0.05, Tukey’s HSD test after one-way ANOVA).

**Figure 4 animals-13-00303-f004:**
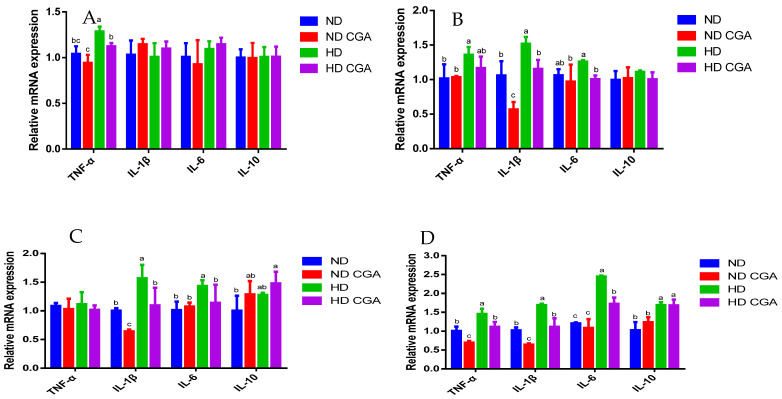
Effects of CGA on jejunal mRNA expression of immune factors in broilers under HD stress. ND group, normal stocking density + basal diet; ND + CGA group, normal stocking density + basal diet + 0.15% CGA; HD group, high stocking density + basal diet; HD + CGA group, high stocking density + basal diet + 0.15% CGA. (**A**) Relative mRNA expression at day 21. (**B**) Relative mRNA expression at day 28. (**C**) Relative mRNA expression at day 35. (**D**) Relative mRNA expression at day 42. Each vertical bar represents the mean ± SEM (n = 10). Values with different letters are significantly different (*p* < 0.05, Tukey’s HSD test after one-way ANOVA).

**Figure 5 animals-13-00303-f005:**
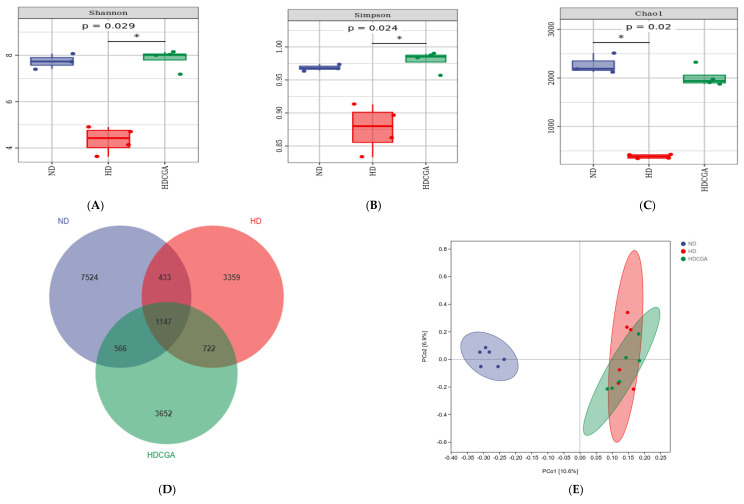
Effect of CGA supplementation on composition and diversity of cecal microbiota of chickens from the ND, HD, and HD + CGA datasets. (**A**–**C**) CGA was found to increase cecal microbial alpha diversity as measured by Shannon, Simpson, and Chao1 indicators. (**D**) Venn diagram showing unique and shared numbers of genera predicted. (**E**) Two-dimensional OTU abundance based principal coordinate analysis (PCoA) of cecal microbiota. (**F**) Microbial composition at the phylum level. (**G**) Microbial composition at the genus level. (**H**) Abundance comparison of major species at the genera level, from left to right were *Faecalibacterium* and *Alistipes*. (**I**,**J**) Leaf and bar plots obtained by linear discriminant analysis effect size (LEfSe) analysis showed differences in the abundance of broiler fecal microbes. (**K**) The clustering heatmap of the top 20 genera in each sample. Green represents positive correlation and yellow indicates negative correlation. “*” presents a significantly different compared to HD group (*p* < 0.05).

**Table 1 animals-13-00303-t001:** Ingredients and nutrient levels in the basal diet.

Ingredient (g/kg)	Starter (1–21 d)	Grower (21–42 d)
Corn	527.9	577.8
Soybean meal	368.9	300.0
Zea gluten meal	0	24.3
Soybean oil	40.0	40.0
Sodium chloride	3.0	3.0
Choline chloride	3.0	2.6
Vitamin premix ^2^	0.3	0.3
Trace element premix ^1^	2.0	2.0
Stone powder	12.2	11.7
Dicalcium phosphate	19.1	16.2
DL-Methionine	2.7	1.1
L-Lysine	0.4	0.45
Wheat bran	20.0	20.0
Total	1000	1000
Metabolic energy (MJ/kg)	12.4	13.0
Crude protein	211.8	198.4
Lysine	11.4	10.5
Methionine	4.9	4.8
Calcium	10.2	8.5
Available P	4.5	4.2
Total P	6.9	6.3
Threonine	7.7	2.2
Analyzed content		
Calcium	10.2	8.5
Total P	6.8	6.2
Calcium: Total P ^3^	1.50	1.37

^1^ Trace element premix is provided as per kg of feed: 8 mg copper (CuSO_4_·5H_2_O); 80 mg iron (FeSO_4_); 100 mg manganese (MnSO_4_·H_2_O); 0.15 mg selenium (Na_2_SeO_3_); 0.35 mg iodine (KI). ^2^ Vitamin premix per kg feed: VA 9500 IU, VD 362.5 µg, VE 30 IU, VK 32.65 mg, VB1 2 mg, VB6 6 mg, VB12 0.25 mg, biotin 325 µg, folic acid 1.25 mg, pantothenic acid 12 mg, niacin 50 mg. ^3^ Calculated nutrient concentrations.

**Table 2 animals-13-00303-t002:** Primer sequences of target genes.

Genes ^1^	Forward Primer (5′-3′)	Reverse Primer (5′-3′)	Length	TM ^2^ °C	Accession No
*OCLN*	ACGGCAGCACCTACCTCAA	GGGCGAAGAAGCAGATGAG	123	51.7	XM_025144247.2
*CLDN1*	CATACTCCTGGGTCTGGTTGGT	GACAGCCA TCCGCA TCTTCT	100	51.3	NM_001013611.2
*CLDN2*	CCTACATTGGTTCAAGCATCGTGA	GATGTCGGGAGGCAGGTTGA	131	50.3	NM_001277622.1
*ZO-1*	CTTCAGGTGTTTCTCTTCCTCCTC	CTGTGGTTTCA TGGCTGGATC	144	51.5	XM_021098886.1
*TNF-α*	GAGCGTTGACTTGGCTGTC	AAGCAACAACCAGCTA TGCAC	176	55.4	NM_214022.1
*IL-1β*	ACTGGGCA TCAAGGGCTA	GGTAGAAGA TGAAGCGGGTC	154	55.6	NM_214005.1
*IL-6*	GCTGCGCTTCTACACAGA	TCCCGTTCTCA TCCA TCTTCTC	203	55.4	NM_204628.1
*IL-10*	AGAAATCCCTCCTCGCCAAT	AAATAGCGAACGGCCCTCA	121	51.2	NM_001004414.2
*GAPDH*	TGCTGCCCAGAACATCATCC	ACGGCAGGTCAGGTCAACAA	142	50–60	NM_204305

^1^ Primer sequences of OCLN, CLDN1, CLDN2, ZO-1, TNF-α, IL-1β, IL-6, IL-10, and GAPDH. ^2^ TM, Melting Temperature.

**Table 3 animals-13-00303-t003:** Effects of CGA on serum antioxidant parameters of broilers under HD stress.

Parameter	Days	Dietary Treatment ^1^				SEM	*p*-Value
		ND	ND + CGA	HD	HD + CGA		
MDA, (nmol/mL)	21	3.56 ^b^	3.52 ^b^	5.19 ^a^	4.04 ^b^	0.22	0.01
	28	4.49 ^b^	4.57 ^b^	5.88 ^a^	4.73 ^b^	0.20	0.03
	35	4.96 ^a^	3.67 ^b^	5.59 ^a^	3.41 ^b^	0.25	<0.01
	42	4.33	4.76	5.54	3.91	0.27	0.16
SOD, (U/mL)	21	148.58	148.13	146.91	147.37	3.55	0.14
	28	162.02 ^a^	154.63 ^ab^	146.70 ^b^	152.83 ^ab^	2.33	0.04
	35	150.43 ^ab^	158.70 ^a^	147.56 ^b^	156.89 ^a^	2.22	0.03
	42	156.96 ^a^	147.24 ^a^	126.91 ^b^	145.82 ^a^	3.92	0.03
GSH-Px, (U/mL)	21	975.78 ^a^	954.36 ^a^	674.99 ^b^	632.35 ^b^	9.44	0.03
	28	911.09 ^b^	936.76 ^ab^	825.41 ^c^	984.84 ^a^	14.75	<0.01
	35	764.49 ^b^	992.03 ^a^	717.75 ^b^	823.66 ^a^	20.92	0.02
	42	910.00 ^ab^	990.15 ^a^	606.69 ^b^	992.31 ^a^	15.92	<0.01
CAT, (U/mL)	21	5.25	5.83	5.27	5.88	0.16	0.33
	28	3.83 ^a^	4.29 ^a^	2.62 ^b^	1.19 ^b^	0.08	<0.01
	35	4.70	5.04	3.45	1.64	0.15	0.58
	42	5.49 ^b^	6.64 ^a^	5.44 ^b^	6.90 ^a^	0.30	<0.01

^1^ ND group, normal stocking density + basal diet; ND + CGA group, normal stocking density + basal diet + 0.15% CGA; HD group, high stocking density + basal diet; HD + CGA group, high stocking density group + basal diet + 0.15% CGA group. Each vertical bar represents the mean ± SEM (n = 10). Values with different letters are significantly different (*p* < 0.05, Tukey’s HSD test after one-way ANOVA).

**Table 4 animals-13-00303-t004:** Effects of CGA on jejunal morphology of broilers under HD stress.

Parameter	Days	Dietary Treatment ^1^				SEM	*p*-Value
		ND	ND + CGA	HD	HD + CGA		
VH (µm)	21	1034.03 ^a^	1023.68 ^a^	852.55 ^b^	946.17 ^ab^	21.43	<0.01
	28	1179.68 ^b^	1290.67 ^a^	976.53 ^c^	1122.33 ^b^	25.38	<0.01
	35	1335.00 ^ab^	1407.33 ^a^	1293.00 ^b^	1366.50 ^ab^	14.11	0.02
	42	1477.33	1450.83	1435.50	1441.50	27.07	0.39
CD (µm)	21	177.83 ^b^	171.77 ^b^	188.10 ^a^	182.37 ^a^	11.10	0.05
	28	179.36 ^a^	173.28 ^b^	181.00 ^a^	175.50 ^ab^	23.13	0.04
	35	171.73	172.61	168.65	167.80	19.13	0.38
	42	167.83 ^b^	167.77 ^b^	174.10 ^a^	173.37 ^a^	20.11	0.05
VH/CD	21	5.73 ^a^	5.67 ^a^	4.56 ^b^	5.07 ^b^	0.13	<0.01
	28	6.74 ^b^	7.71 ^a^	5.74 ^c^	6.39 ^b^	0.18	<0.01
	35	7.78	8.16	7.67	8.15	0.09	0.13
	42	8.81 ^a^	8.65 ^a^	8.24 ^b^	8.31 ^ab^	0.08	0.02

^1^ ND group, normal stocking density + basal diet; ND + CGA group, normal stocking density + basal diet + 0.15% CGA; HD group, high stocking density + basal diet; HD + CGA group, high stocking density group + basal diet + 0.15% CGA group. Each vertical bar represents the mean ± SEM (n = 10). Values with different letters are significantly different (*p <* 0.05, Tukey’s HSD test after one-way ANOVA).

## Data Availability

The corresponding author can provide access to the collected and analyzed data sets from this study upon request.

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
