# Peer review of "Dietary Supplementation with Chlorogenic Acid Enhances Antioxidant Capacity, Which Promotes Growth, Jejunum Barrier Function, and Cecum Microbiota in Broilers under High Stocking Density Stress"

_animals, 2023, doi:10.3390/ani13020303_

Round 1

Reviewer 1 Report

Comments to the authors

This study reported that effects of the dietary supplementation with chlorogenic acid enhances antioxidant capacity, which promotes growth, jejunum barrier function, and caecum microbiota in broilers under high stocking density stress. 

Major Comments

1.     The all data must be re-analyzed by using tukey’s multiple test or other more appropriate methods, because Duncan's method is not recommended. If these data are re-analyzed by other statistical method, the interpretation of the results from this study may be changed.

2.     How pure is the GCA used in this study? Also, how did authors decide the amount of GCA added? In addition to materials and methods, the reviewer considers that it is better to explain.

3.     This manuscript must be checked by professional English editing service. 

Author Response

Dear Reviewer

Thank you for your suggestions and comments. We have revised the manuscript and added the documents according to your comments.

Reviewer 2 Report

Major comments:

The raw sequence data needs to be uploaded to a public repository. Provide the accession number in the manuscript.

2.6: Detailed protocols for RNA-extraction, reverse transcription and qPCRs are missing.

2.7: PCR conditions for amplicon generation and protocol for library generation are missing. More information about quality filtering, OTU clustering, etc. should be provided.

Figures 1, 3, and 4: Bar plots can be misleading because they reveal little about the distribution of data. These data should be presented as boxplots instead.

Figure 5A-C: What is the difference between the bar and the boxplots? The y-axis in the barplots of Shannon and Simpson index should start at 0.

Figure 5K and lines 348-352: Picrust2 is a controversial software that tries to predict metabolic functions based on 16S rRNA gene distributions. While this is possible to some extent, it has many flaws and could lead to wrong conclusion in some cases (e.g., Species with identical 16S but different metabolisms). I don´t see how the Picrust2 analysis adds any additional value to the study and I would advise the authors to consider removing it completely. They provide enough data and interesting results that warrant publication without the additional Picrust2 analysis. However, it is not wrong per se to perform such an analysis. So, if the authors decide to keep it, they should interpret the results with care (See my comments on lines 486-500).

Figure 5J: I don´t really understand what this heatmap shows or how it was created (This information is missing in the methods section). The legend states that it shows the correlation of SCFAs and the top 20 genera in each group. It is not clear what SCFA is.

Line 486-500: Picrust2 predicts metabolic functions based on 16S rRNA gene distributions, which is highly speculative. Results should be interpreted with care. Please rephrase or delete statements like this: “In this study, through COG functional classification, it was found that intestinal bacteria in the HD + CGA group had higher metabolic activity than in the HD group”. Metabolic activity was not measured, it was just predicted to be higher, but whether that is actually the case was not tested.

The discussion section is too broad and should be condensed to more precisely define the scope of the study.

Specific comments:

Line 22: …the impaired what?

Line 28: What is SOD? Abbreviations need to be specified when first used.

Line 71: microbial population

Line 210: Is this correct? In Figure 1B, ND and HD + CGA are not significantly different.

Line 227: SOD and GSH-Px are not significantly lower in HD compared to ND in table 3.

Line 231: Again, the text does not match with the results presented in table 3

Line 244-246: Text does not match with the results presented in table 4

Line 291: What do you mean by single species indices?

Line 298: shared not duplicated OTUs

Line 310-313: How were these comparisons calculated?

Line 316 and 319: What do you mean by colony members and biomarkers? Rephrase

Figure 5H-I: Lefse can perform comparisons of 3 groups. Why did you choose to compare 2 groups separately and why were the chosen thresholds different?

Line 367: What is MAD? Do you mean MDA?

Line 372: Again MAD

Line 378 and 409: What is DEX? Abbreviations need to be specified when first used.

Line 393-394: Is this statement correct? All the studies you cite and your own are in agreement.

Line 447: microbial population

Line 455-457: I disagree. The PCoA did not show that the HD + CGA group samples had a similar distribution in population structure. Certain taxa changed in abundance but the overall structure of the population does not seem to be very different from the HD group.

Line 465: Delete “.” after microbiota

Author Response

Dear Reviewer

Thank you for your suggestions and comments.

We have revised the manuscript and added the documents according to your comments

Round 2

Reviewer 2 Report

Line 179: was isolated is written twice

Line 208: Please add detailed parameters under which flash and trimmomatic were run.